# Physical Activity in Older Adults: An Investigation in a Metropolitan Area of Southern Italy

**DOI:** 10.3390/ijerph17031034

**Published:** 2020-02-06

**Authors:** Francesca Gallè, Elita Anna Sabella, Giovanna Da Molin, Eduardo Alfonso Parisi, Giorgio Liguori, Maria Teresa Montagna, Osvalda De Giglio, Luca Tondini, Giovanni Battista Orsi, Christian Napoli

**Affiliations:** 1Department of Movement Sciences and Wellbeing, University of Naples “Parthenope”, Via Medina n. 40, 80133 Naples, Italy; giorgio.liguori@uniparthenope.it (G.L.); l.tondini@libero.it (L.T.); 2Inter-University Research Centre “Population, Environment and Health”, University of Bari Aldo Moro, Piazza Umberto I, 1, 70121 Bari, Italy; elita.sabella@uniba.it (E.A.S.); giovanna.damolin@uniba.it (G.D.M.); 3Department of Medical Surgical Sciences and Translational Medicine, “Sapienza” University of Rome, Via di Grottarossa 1035/1039, 00189 Rome, Italy; eduardoparisiroma@gmail.com (E.A.P.); christian.napoli@uniroma1.it (C.N.); 4Department of Biomedical Science and Human Oncology, University of Bari Aldo Moro, Piazza G. Cesare 11, 70124 Bari, Italy; igiene.dimo@uniba.it (M.T.M.); osvalda.degiglio@uniba.it (O.D.G.); 5Department of Public Health and Infectious Diseases, “Sapienza” University of Rome, Piazzale Aldo Moro 5, 00185 Rome, Italy; giovanni.orsi@uniroma1.it

**Keywords:** inactivity, elderly, sociodemographic determinants

## Abstract

Physical activity (PA) and exercise are fundamental to maintaining health in older adults. World Health Organization guidelines state that older adults should practice at least 150 min/week of moderate/vigorous intensity PA to obtain health benefits. We assessed PA levels among older adults in southern Italy and evaluated possible associated determinants. The study was carried out between September and November 2019 in the metropolitan area of Bari. We collected information from participants over 65 years using a self-administered questionnaire. We investigated associations between sociodemographic characteristics, health conditions, and inactivity/PA levels. A total of 383 individuals completed the questionnaire. Mean body mass index indicated that 45.4% of participants were overweight. Mean time spent in PA was 476.2 ± 297.8 min/week, with women reporting lower levels than men (370.8 ± 210 vs. 555.2 ± 334.3 min/week, *p* = 0.08). Weekly sitting time was positively related to age. Attending religious or recreational activities was related to moderate PA. Educational level was positively associated with PA while dog ownership represented an obstacle to achieving recommended PA levels in our population. Participants generally met the recommended levels of PA, especially men; the educational level was the main determinant. Interventions aimed at promoting PA among older adults with lower education levels and women are needed in this setting.

## 1. Introduction

The contribution of physical inactivity to the development of chronic health conditions is widely recognized, and evidence shows that this risk increases with age [1,2]. Ageing is commonly associated with a decline in cardiorespiratory fitness, muscle strength, agility, and balance, as well as in cognitive performance, which together can lead to reduced abilities and independence [3]. The onset of other factors related to physical inactivity, such as high blood pressure, high glucose levels, and overweight/obesity, is also common in older adults, leading to an increased risk of mortality and lower quality of life [4,5]. In contrast, physical activity (PA) and exercise have been proven as fundamental in maintaining health resources in older people [6,7]. Therefore, several international and national institutions as well as governments have published recommendations that older people should engage in PA. The World Health Organization (WHO) guidelines state that older adults should practice at least 150 min per week of moderate-to-vigorous intensity PA to obtain health benefits [8]. In Italy, a recent evidence-based publication of the Italian Ministry of Health states that older adults can achieve the recommended PA levels through the adoption of an active lifestyle, which may include active transportation, such as walking, working, and leisure activities that include planned exercise in all their life settings [9]. Nevertheless, the surveillance system of the Italian Institute of Statistics reports that individuals aged 65–74 years (47.3%) and > 75 years (71.3%) have the highest inactivity rates in the Italian population, suggesting the need for a better understanding of the determinants associated with inactivity to then promote PA interventions targeting these population groups [10]. 

Previous scientific evidence has shown that some sociodemographic, psychological, physical, and behavioral factors may represent barriers to PA and exercise in older adults. Sex, age, marital status, educational level, income, health status, social participation, past engagement in exercise/sports, caring for grandchildren, and dog ownership have been identified in previous investigations as determinants of practicing PA in older adults [11,12,13,14]. In particular, moderate evidence has been found regarding the negative association between female sex and PA, as well as older age and PA [13]. The characterization of these factors is essential for planning policies of PA promotion that are adapted to different geographical contexts.

To our knowledge, no data are available concerning PA among older adults in the metropolitan area of Bari, a city in southern Italy. We hypothesized that PA in older residents of Bari is low and that some previously identified determinants are associated with these low PA levels. Therefore, the aim of this cross-sectional study was to assess PA levels among older adults in Bari and to evaluate the association with sociodemographic characteristics, health conditions, and behaviors to identify possible determinants for future PA promotion strategies.

## 2. Materials and Methods 

### 2.1. Participants and Setting

This study was carried out between September and November 2019 in the metropolitan area of Bari (MAB) among a sample of older adults. The inclusion criteria were being a resident of MAB and age ≥65 years. We excluded individuals with psychological or physical conditions limiting their physical abilities.

The MAB is divided into five administrative districts. According to the latest census in 2016, the MAB had a resident population of 75,483 older adults [15]. Participants were recruited by approaching older adults attending church/religious, recreational, and cultural activities in each district of the MAB and asking them to involve other relatives and acquaintances. 

Using a quota sampling method, assuming a 95% confidence level and a 5% margin of error, the sample was stratified according to the percentage of older residents in each administrative district [13] and to the distribution by sex of the total population (32,334 men and 43,149 women). A final sample of 383 individuals (mean age 73.2 ± 6.7, 42.8% men) was recruited. 

The investigation was performed in accordance with the World Medical Association Declaration of Helsinki. This study did not include any experiments involving human or biological human samples, nor research on identifiable human data. According to Italian laws, the research was conducted anonymously.

### 2.2. Questionnaire

Data were collected using a self-administered questionnaire. The questionnaire included a first part regarding demographic characteristics, such as sex, age, administrative district of residence, nationality (Italian/other), educational level (none/elementary/middle/high/university degree), marital status (never married/married/cohabitating/separated/divorced/widowed), living situation (alone/with spouse/with spouse and children/with children/with family of origin/with other cohabitants/other), occupational status (retired or not retired), and type of work performed (physical/non-physical). Interviewees were also asked to self-report their anthropometric measures (height and weight values) so as to calculate their body mass index (BMI) and related weight status (underweight/normal weight/overweight/obese), according to the WHO classification [16].

The second part of the questionnaire aimed to collect information regarding the social relationships that are considered indicators of an active lifestyle, i.e., caring for grandchildren (e.g., accompanying grandchildren to school, playground, or recreational facilities by active transport at least 3–4 times per week), dog ownership (walking a dog daily), and attendance (at least twice per week) at social gatherings, such as church/religious, cultural, or recreational activities. Self-reported health conditions were also queried (chronic condition/disease).

The third part of the questionnaire investigated participants’ PA levels in questions related to their past by asking them to report previous engagement in sports/PA, as well as their current practice of moderate PA (days per week and minutes per day), vigorous PA (days per week and minutes per day), walking (days per week and minutes per day), going to a gym (yes/no), and sedentary lifestyle (average weekly time spent sitting). These variables were investigated through administration of the International Physical Activity Questionnaire [17].

A brief introduction was included in the questionnaire to explain the aims of the study, the treatment of data, and the definitions of moderate and vigorous PA. Participants expressed their informed consent to participate in the investigation by completing the questionnaire.

### 2.3. Statistical Analyses

Descriptive analysis was performed considering sociodemographic characteristics and PA-related habits of the entire sample and of subgroups by sex. Continuous outcomes were expressed as the mean ± standard deviation (SD). The Student *t*-test was used to compare mean values between the two sexes. Data regarding conditions and behaviors were reported as the number and percentage of respondents; these variables were compared between males and females using the chi-squared test. 

Linear regression analyses were carried out to detect possible relationships between sociodemographic characteristics (age, sex, nationality, educational level, retirement, marital status, living situation, care of grandchildren/dogs, and attendance at church/religious, cultural, or recreational activities), health conditions (BMI, chronic conditions/diseases), previous sports/PA practice, type of work, and current inactivity/PA levels (weekly sitting time, weekly time spent walking, moderate or vigorous activity, total PA). All these outcomes were investigated separately using forward selection. Beta values and 95% confidence intervals (CIs) for predictors were included in the best fitting models. Adjusted R^2^ values were reported for each model. 

Multinomial logistic regression analysis was performed to evaluate the possible association between the abovementioned variables and the achievement of PA levels recommended by the WHO for older adults. The outcome was established by attributing a value of 1 if the participant reported at least 150 min of moderate/vigorous activity per week, and a value of 0 otherwise. Odds ratios (ORs) and 95% CIs for all predictors were calculated. A value of *p* < 0.05 was assumed as the level of significance. Data were analyzed using IBM SPSS version 25 for Windows (IBM Corp., Armonk, NY, USA). 

## 3. Results

A total of 383 individuals participated in this study by completing the questionnaire. The main characteristics of the sample are shown in Table 1. In general, the study population mainly comprised Italian retirees with a high educational level; most were married and lived with their spouses, did not care for their grandchildren or own a dog, and attended social gatherings, especially in local parishes. The mean BMI indicated a high prevalence of overweight, and about 75% of the sample had a chronic condition or disease. Most respondents reported previous exercise (mainly PA) and non-physical work in the past. 

As for sex differences, men had a significantly higher educational level than women; men also represented a higher proportion of retired and married people, living mainly with their wives. The mean BMI in men was also higher than that in their female counterparts. Regular attendance at social gatherings was mainly reported by women, as was previous non-physical work and continuous PA throughout their lifetime.

Table 2 shows the characteristics of participants related to the practice of sports/PA. Mean levels of inactivity and PA showed that our study population generally met the recommended levels of PA for health. However, only a few participants reported going to a gym. Men reported higher levels of current PA than women, even though only the difference in moderate PA was significant. The proportion of men who achieved the recommended levels of moderate/vigorous PA was significantly higher than that of their counterparts, although women reported going to a gym more often than men.

Table 3 shows the best fitting linear regression models for each variable related to inactivity/PA. Weekly sitting time was positively related to age and negatively associated with previous practice of sports/PA. Previous sport/PA seemed to also positively influence the current amount of walking, together with educational level and dog ownership. Attending church/religious or recreational activities and having a chronic condition or a disease were related to moderate PA levels whereas vigorous PA was positively related to educational level. The total weekly PA level was associated with health condition.

The results of the multinomial logistic regression, considering attaining the recommended weekly PA levels as the outcome, are shown in Table 4. All independent variables that were significantly associated with the outcome in the Pearson’s correlation test were included in the analysis. Being male and married, having a degree, attending church/religious and recreational activities, having a higher BMI and a chronic health condition, and reporting previous physical work and previous sports/PA had a positive role in reaching the recommended PA levels. Being older and not Italian, retired, living with others, taking care of grandchildren or owing a dog, and attending cultural activities represented obstacles to achieving the recommended PA levels. However, only the associations with educational level and dog ownership were significant.

## 4. Discussion

This study reports the results of a survey investigating PA levels and determinants in a sample of older adults selected from the metropolitan area of a large city in southern Italy. 

The reported levels of weekly PA met the recommendations for health in older people [8]. This is in contrast with previous reports [18,19] as well as with the data reported by the Italian Institute of Statistics, which show inactivity levels of nearly 50% among populations of southern Italy, with only about 20% of older adults practicing continuous PA [10]. Moreover, as previously reported, even young adults from the MAB show a high level of inactivity (> 65%), with no differences by sex [20]. The characteristics of our sample (mainly comprising Italian, retired, highly educated individuals with previous non-physical work and previous exercise, who regularly attended social gatherings) may explain our findings.

More men than women in our sample reported significantly higher levels of moderate PA and achieved recommended PA levels. These data confirm the sex differences among older adults reported in the literature [13]. However, our findings are not completely consistent with those of a previous population-based cohort study reporting that women had greater levels of light PA but lower levels of sedentary behavior and moderate-to-vigorous PA than men [21]. A systematic review analyzing the effects of PA levels on the activities of daily living in older adults found the largest health effects for moderate PA, concluding that moderate PA levels may produce the greatest benefits in performing the activities of daily living [22]. Therefore, the lower levels of moderate PA reported among women may have important consequences for their independence. 

Regarding the roles of the other variables examined in this study, older age was found to be a determinant of inactivity. The previous practice of sports/PA might prevent higher current levels of sitting time in favor of current walking habits, as reported in the literature [23].

A higher educational level was identified as a determinant for walking and vigorous PA, and this was also positively associated with achievement of the recommended PA levels. In fact, a previous study showed that a low educational level is a determining factor in declining leisure-related PA [24] whereas a higher educational level is associated with greater adherence to community exercise programs [25]. Moreover, although common in individuals aged 50 years or more who were recently diagnosed with a health condition, it has been shown that behavioral changes, such as starting PA, are more common in more highly educated individuals [26]. This finding is in line with the positive association between chronic conditions and moderate or total PA found in our study. It is known that people with chronic conditions are considerably less physically active, even though PA in individuals with chronic conditions may improve associated depression, mobility difficulties, and pain, which are important barriers to initiating or adopting an active lifestyle [27]. Considering that about three-quarters of our sample had a chronic condition or disease, it is possible that the higher education levels among participants may have played a role in determining a higher compliance with their physicians’ recommendations regarding an active lifestyle to better manage their health condition.

Furthermore, moderate and total PA were positively associated with regular attendance at church/religious or recreational activities. It is well known that social relationships are central to the health and well-being of aging populations and that social isolation is associated with sedentary behavior [28]. Therefore, our findings confirm this association.

Owning a dog was a determinant of walking in our sample. This is in line with previous studies reporting an association between dog ownership and walking [29,30]. However, contrary to the published literature, achievement of the recommended PA levels was inversely related to dog ownership [31]. It is possible that in this age category, owing a dog may hinder the practice of higher-intensity PA in favor of walking. Further studies should investigate this issue.

Most of our participants were overweight or obese, confirming international evidence that the percentage of older overweight/obese adults is on the rise [32]. However, as shown in the results of the regression analyses, BMI was not associated with inactivity nor with PA levels in our study.

This study has some limitations. First, the outcomes considered were all self-reported and not objectively measured. Therefore, it is possible that some of the information reported, especially regarding weight and levels of inactivity/PA, was inaccurate. Furthermore, the sample selection may have been biased by the recruitment of a greater proportion of highly educated individuals with previous non-physical occupations and who engaged in PA throughout their lives. In this regard, comparison with a sample from a rural area may have been useful to enhance our findings.

Nevertheless, the sample size and its representativeness of the older population in the MAB are the main strengths of this study. Furthermore, it should be noted that in contrast to infectious diseases, where surveillance systems may help with early detection, continuous surveillance systems for risk factors for non-transmittable diseases, such as overweight and physical inactivity, are very difficult to implement and maintain. However, this type of epidemiological monitoring may be useful to establish timely and more effective interventions [33,34].

## 5. Conclusions

In this study, older adults living in the metropolitan area of Bari, Italy showed mean levels of PA that met the recommended levels of PA for health in older individuals [8,9]. Our data were not consistent with national data reporting higher inactivity rates in the Italian population [10]. As reported in other countries, men showed higher PA levels than women and educational level was the main determinant. 

The main findings in this study highlight the need for health promotion interventions targeting specific population subgroups and focusing on specific issues. Promoting PA among older women and individuals with lower educational levels may help to reduce inactivity and related health issues in this setting.

## Figures and Tables

**Table 1 ijerph-17-01034-t001:** Participant characteristics and *p* values for comparison between groups, by sex.

	Participants, N = 383			
	Total	Men n = 164	Women n = 219	*p*
Age (y), mean ± SD (range)	73.2 ± 6.7	73.7 ± 6.9	72.8 ± 6.5	0.23 ^a^
	(65–94)	(65–94)	(65–94)	
Nationality, n (%)				0.47 ^b^
Italian	379 (99)	163 (99.4)	216 (98.6)	
Other	4 (1)	1 (0.6)	3 (1.4)	
Educational level, n (%)	15 (3.9)	3 (1.8)	12 (5.5)	0.02 ^b^
None	32 (8.4)	15 (9.2)	17 (7.8)	
Elementary	103 (26.9)	33 (20.2)	70 (32.1)	
Middle school	137 (35.8)	66 (40.5)	71 (32.6)	
High school degree	94 (24.5)	46 (28.2)	48 (22)	
Retired, n (%)	305 (79.6)	141 (86)	164 (75.9)	0.015 ^b^
Marital status, n (%)				0.004 ^b^
Never married	27 (7)	9 (5.5)	18 (8.2)	
Married	226 (59)	110 (67.1)	116 (53)	
Cohabitating	0	0	0	
Separated/divorced	30 (7.8)	16 (9.8)	14 (6.4)	
Widower/widow	100 (26.2)	29 (17.7)	71 (32.4)	
Living alone, n (%)	100 (26.1)	35 (21.3)	65 (29.7)	0.009 ^b^
Living with, n (%)				
Husband/wife	149 (38.9)	73 (44.5)	76 (34.7)	
Spouse and children	77 (20.1)	38 (23.2)	39 (17.8)	
Children	31 (8.1)	11 (6.7)	20 (9.1)	
Family of origin	9 (2.3)	1 (0.6)	8 (3.7)	
Other cohabitants	10 (2.6)	6 (3.7)	4 (1.8)	
Other	7 (1.8)	0	7 (3.2)	
Taking care of grandchildren, n (%)	163 (42.6)	67 (40.9)	96 (43.8)	0.56 ^b^
Dog owner, n (%)	91 (23.8)	44 (26.8)	47 (21.5)	0.22 ^b^
Regularly attending social gatherings, n (%)	280 (73.1)	109 (66.5)	171 (78.1)	0.01 ^b^
Church or religious gathering	212 (55.4)	78 (47.6)	134 (61.2)	0.008 ^b^
Cultural activities	76 (19.8)	27 (16.5)	49 (22.4)	0.15 ^b^
Recreational activities	72 (18.8)	35 (21.3)	37 (16.9)	0.27 ^b^
Body mass index,	27.2 ± 4.1	27.8 ± 3.3	26.7 ± 4.5	
mean ± SD (range)	(15.8–41)	(20.15–38.78)	(15.78–41.01)	0.01 ^a^
Underweight	4 (1)	0	4 (1)	
Normal weight	114 (29.8)	33 (20.1)	81 (37)	
Overweight	174 (45.4)	86 (52.4)	88 (40.2)	<0.01 ^b^
Obese	91 (23.8)	45 (27.4)	46 (21)	
At least one health chronic condition/disease	332 (86.7)	142 (86.6)	190 (86.8)	0.96 ^b^
Previous physical work, n (%)	29 (8)	23 (14.5)	6 (2.9)	<0.01 ^b^
Previous sports/PA, n (%)				
Continuous PA	100 (26.1)	30 (18.3)	70 (32)	0.02 ^b^
Recreational sports	57 (14.9)	30 (18.3)	27 (12.3)	
Amateur sports	44 (11.5)	22 (13.4)	22 (10)	
Competitive sports	18 (4.7)	10 (6.1)	8 (3.7)	

^a^*t*-test; ^b^ Chi-squared test; Abbreviations: PA, physical activity; SD, standard deviation.

**Table 2 ijerph-17-01034-t002:** Current levels of inactivity and PA reported by participants and *p* values for the comparison between groups, by sex.

	Participants (N = 383)			
	Total	Men n = 164	Women n = 219	*p*
Sitting (min/week),	1254.9 ± 885.2	1253.1 ± 882.5	1256.2 ± 889.6	0.97 ^a^
mean ± SD (range)	(35–5040)	(35–4200)	(70–5040)	
Walking (min/week),	208.8 ± 136.9	222.7 ± 144.3	198 ± 130.2	0.11 ^a^
mean ± SD (range)	(10–630)	(10–630)	(20–630)	
Moderate activity (min/week),	133.8 ± 90	159.2 ± 98.4	114.7 ± 78.5	<0.01 ^a^
mean ± SD (range)	(10–420)	(30–420)	(10–420)	
Vigorous activity (min/week),	117.6 ± 118.3	136.3 ± 152	97.3 ± 61.9	0.28 ^a^
mean ± SD (range)	(5–720)	(5–720)	(5–180)	
Total PA (min/week),	476.2 ± 297.8	555.2 ± 334.3	370.8 ± 210	0.08 ^a^
mean ± SD (range)	(63–1050)	(63–1050)	(120–950)	
Meeting recommended PA level, n (%)	161 (42)	92 (56.1)	69 (31.5)	0.01 ^a^
Going to the gym, n (%)	103 (26.9)	38 (23.2)	65 (29.7)	0.15 ^b^

^a^*t*-test; ^b^ Chi-squared test; Abbreviations: PA, physical activity; SD, standard deviation.

**Table 3 ijerph-17-01034-t003:** Linear regression models with the corresponding values of adjusted R^2^ for variables related to PA and inactivity reported by participants.

	Sitting	Walking	Moderate Activity	Vigorous Activity	Total PA
			Beta values (95% CI) *p*		
Age	20.1 (5.5–34.6) **	0.44 (−1.6–2.5)	−0.85 (−2.1–0.37)	−0.28 (−1–0.5)	−0.7 (−3.4–2)
Sex	102.7 (−103.1–308.7)	−19.7 (−51.9–12.5)	−14.1 (−33.1–4.8)	−11.5 (−23.5–0.37)	−45.4 (−88.4–−2.4)
Nationality	−556.2 (−1358.3–245.9)	10.5 (−118.8–139.9)	102.6 (26.4–178.8)	59.8 (11.9–107.8)	173 (0.49–345.6)
Educational level	−41.1 (−63.7–145.9)	51.1 (19.8–82.3) **	0.35 (−9.8–9.1)	30.3 (21.3–39.3) **	21.6 (0.15–43.1)
Retirement	197.8 (−474.9–79.4)	11.7 (−30.7–54)	−3.4 (−28.4–21.5)	5.7 (−9.9–21.4)	13.9 (−42.5–70.4)
Marital status	28.3 (−78.2–134.7)	−10.5 (−27.3–6.3)	0.92 (−8.9–10.8)	6.3 (0.05–12.5)	−3.2 (−25.7–19.2)
Living conditions	−19.3 (−88.5–50)	−9.6 (−20.4–1.1)	−1.3 (−7.6–5.1)	2.1 (−6.1–1.8)	−13 (−27.4–1.3)
Caring for grandchildren	113.6 (−83.8–311)	25.2 (−5.6–56)	−1.1 (−19.3–17)	−2.9 (−14.3–8.5)	21.1 (−19.9–62.3)
Dog owner	−79.4 (−152.3–311.2)	106.6 (36.7–176.4) **	3.8 (−17.3–24.8)	−8.9 (−22.1–4.3)	34.6 (−13–82.3)
Attending church	−181.9 (−389.3–25.3)	37.4 (5.2–69.7)	29.8 (2.1–57.6) *	7 (−4.9–18.9)	38.1 (−4.8–81.1)
Attending cultural activities	22.9 (−238.5–284.4)	−21.8 (−62.9–19.2)	16.7 (−7.5–40.9)	6.5 (−8.7–21.7)	1.4 (−53.3–56.1)
Attending recreational activities	96.9 (−159.9–353.8)	−58.1 (−98–−18.2)	39.2 (8.48–69.9) **	−12 (−26.8–2.7)	−116 (−169.3–−62.8)
BMI	2.2 (−16.6–20.9)	1.8 (−1.1–4.8)	1.5 (−0.2–3.3)	0.4 (−0.6–1.5)	3.8 (−0.1–7.8)
Health chronic condition/disease	145.8 (−166.4–457.9)	33.7 (−13.2–80.7)	57.4 (137–101.1)**	−18.3 (−35.7–−0.9)	269.3 (204.7–334) **
Previous physical work	−42.4 (−428.9–344.1)	−27.4 (−86.3–31.4)	−7.2 (−41.9–27.4)	3.3 (−18.5–25.1)	−31.4 (−109.9–47.1)
Previous sport/PA	−203.8 (−400–−7.5) *	159 (72.3–245.7) **	−23.3 (−43–−3.6)	−17.7 (−30.1–−5.3)	−87.2 (−131.8–−42.5)
Adjusted R^2^	0.67	0.59	0.34	0.11	0.63

* *p* < 0.05. ** *p* < 0.01; Abbreviations: PA, physical activity; CI, confidence interval; BMI, body mass index.

**Table 4 ijerph-17-01034-t004:** Results of the multinomial logistic regression for the achievement of recommended weekly PA level as the outcome.

	Meeting Recommended PA Level	
	OR (95% CI)	*p*
Age group		
65–74 years	*reference*	
>74 years	0.8 (0.05–13.9)	0.88
Sex		
Female	*reference*	
Male	7.9 (0.5–126.7)	0.14
Nationality		
Italian	*reference*	
Other	0.6 (0.04–8.2)	0.67
Educational level		
No degree	*reference*	
Degree	52.8 (1.4–1950.7)	0.03
Retired		
No	*reference*	
Yes	0.6 (0.2–1.6)	0.35
Marital status		
Unmarried	*reference*	
Married	2 (0.6–7.4)	0.25
Living situation		
Alone	*reference*	
Living with others	0.2 (0.04–1.4)	0.11
Caring for grandchildren		
No	*reference*	
Yes	0.11 (0.01–2.1)	0.14
Dog owner		
No	*reference*	
Yes	0.03 (0.01–0.7)	0.03
Attends church/religious association		
No	*reference*	
Yes	1.9 (0.9–3.8)	0.07
Cultural activities		
No	*reference*	
Yes	0.5 (0.2–1.1)	0.09
Attends recreational activities		
No	*reference*	
Yes	29.4 (0.28–3030.2)	0.15
BMI		
Underweight/normal weight	*reference*	
Overweight/obese	2 (0.4–4.9)	0.11
Chronic health condition/disease		
No	*reference*	
Yes	1.2 (0.4–3)	0.72
Previous physical work		
No	*reference*	
Yes	2.3 (0.6–9.1)	0.23
Previous sports/PA		
No	*reference*	
Yes	3.2 (0.01–2575.4)	0.72

Abbreviations: PA, physical activity; OR, odds ratio; CI, confidence interval; BMI, body mass index.

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
