# Peer review of "Physical Activity in Older Adults: An Investigation in a Metropolitan Area of Southern Italy"

_ijerph, 2020, doi:10.3390/ijerph17031034_

Round 1

Reviewer 1 Report

please see the attached file.

Author Response

Reviewer 1

We are grateful to Reviewer 1 for his kind suggestions.

We have tried to fulfill his indications and we hope that the quality of the manuscript has improved.

Abstract

Line 21, “Background”, not “Introduction”

Line 24, “was”, not “is

Please use abbreviation of physical activity-PA

Line 25, “Methods”, not “Materials and Methods”

Line 27, “self-administrated questionnaires”, not “self-completed questionnaires”.

“Possible relationship” – “Associations”

Line 28, how about the mean and SD of the 383 older adults? I suggest the authors report the extent to which older adults reach PA guidelines of 150 mins moderate-to-vigorous PA.

Line 32, “have a degree plays a positive role in reaching recommended PA levels” --- “educational levels positively associated with”

The keywords listed within the manuscript are pulled from the study title. Keywords are generally utilized to assist search engines in locating studies. Suggested replacing keywords that are located in the title of the manuscript.

All the changes and integrations suggested were made. Different keywords have been chosen.

Introduction

There are four paragraphs in the Introduction section. The logic flow of introduction is weak. The authors should provide a clear rationale behind the study purposes. In the models (Table 3 and Table 4), the authors examined the associations between PA and several factors (i.e., age, educational level, retirement, dog, parish, recreational, diseases, past PA behaviours). Why the authors included these variables in the models? Is there any theoretical support or empirical evidence? What is the research gap? Please provide study hypotheses and covariates at the end of the Introduction section. Both of them should have citations.

We thank the referee comment that allowed to improve the introduction section:

we presented data concerning the inactivity in the elderly as a public health issue, followed by international and national recommendation we reported the situation of PA in Italy, characterized by highest inactivity rates in the Italian older population, suggesting the needs of better understand the determinants we reported, according to scientific literature, the factors associated to physical inactivity (adding some of the main reference in this field) considered that, to our knowledge, no data are available concerning PA of older adults in the metropolitan area of Bari, our hypothesis is that also in this population, PA in older people is low and that some determinants already identified in previous studies can be associated to this habit. Therefore, the study was aimed to assess the PA level of older adults in this setting and to evaluate its association to possible determinants for future PA promotion strategies.

Methods

Line 68, delete “to be included in the study”, the authors should provide more information about the inclusion and exclusion criteria.

We have changed the sentence by specifying inclusion and exclusion criteria through the following sentence:

The inclusion criteria were being a resident of MAB and age ≥65 years. We excluded individuals with psychological or physical conditions limiting their physical abilities.

How about age (mean and SD) and gender (%female) of the sample?

Age and gender have been added to the description of the sample in this section.

Line 78, what is anonymous self-completed questionnaire?

The corresponding sentence has been changed as follows: “Data were collected using a self-administered questionnaire.”

Line 80, educational levels, what are the middle and high degrees?

We agree, the right expression should have been “(none/elementary/middle/high/university degree)”. It has been corrected in the text.

Lines 87-91, please provide citations of the measurements if the measures are not developed by the authors.

The questions related to the social participation were developed considering the possible activities that may contribute to the achievement of the recommended PA levels. We have detailed this as follows: “The second part of the questionnaire aimed to collect information regarding the social relationships that are considered indicators of an active lifestyle, i.e., caring for grandchildren (e.g., accompanying grandchildren to school, playground or recreational facilities by active transport at least 3–4 times per week), dog ownership (walking a dog daily), attendance (at least twice per week) at social gatherings such as church/religious, cultural, or recreational activities. Self-reported health conditions were also queried (chronic condition/disease).”

Lines 120-123, ethics information should not be in the Data analysis part.

If the process of data collection is anonymous, how did the participants sign the consent form?

Participants did not sign a written informed consent. The introduction of the questionnaire stated that by accepting to complete the questionnaire, the participants would have expressed their informed consent to take part to the investigation. A corresponding sentence has been added to the “Questionnaire” section.

Results

Table2, Is there difference in the proportion of older adults reaching PA guidelines?

The number and proportion of subjects reaching recommended PA levels on the total sample and divided per gender has been included in Table 2.

Did the authors include any covariates in the linear and logistic models? If NO, why? If YES, please list the covariates at the end of Table 3 and Table4

Covariates were not used because we considered all the variables as possible determinants of PA and we wanted to analyze their role as predictors. As stated in the introduction, literature shows that age and gender were proven to be determinants of PA in older adults. Therefore, we included also these factors in the regression analyses.

In Table3, the authors should not only report sig findings.

Table 3 and Table 4 were integrated with all the results. We have also added the values of R2 for each linear regression model.

Table 4 is difficult to read. The authors should put P-value in a separate column.

All the independent variables were added and p values were reported in a separate column.

Discussions

Line 221, the authors did not report any result about the association between BMI and PA.

Tables 3 and 4 have been integrated with the non-significant results. Therefore, the lack of significant associations between BMI and the different outcomes has been shown.

We have changed the corresponding sentence in the Discussion as follows:

“However, as shown by the results of the regression analyses, in our study BMI was not associated with inactivity nor with PA levels.”

Conclusions

The authors should improve the Conclusion section. What is “generally fulfill PA guidelines”?

The Conclusions section has been changed. The first two sentences have been rephrased as follows: “In this study, older adults living in the metropolitan area of Bari, Italy showed mean levels of PA that met the recommended levels of PA for health in older individuals [8,9]. Our data were not consistent with national data reporting higher inactivity rates in the Italian population [10]. As reported in other countries, men showed higher PA levels than women and educational level was the main determinant.”

Proofreading by native speakers is recommended.

A native English speaker has reviewed the language of the manuscript.

Reviewer 2 Report

This study aimed to examine the levels of physical activity in old adults of a metropolitan area of Southern Italy. However, this study is limited by its cross-sectional design, using self-report PA measures, a number of grammar mistakes, and not adding novel findings in this area. 

Author Response

Reviewer 2

This study aimed to examine the levels of physical activity in old adults of a metropolitan area of Southern Italy. However, this study is limited by its cross-sectional design, using self-report PA measures, a number of grammar mistakes, and not adding novel findings in this area. 

We are sorry for the negative judgement of Reviewer 2. Also thanks to the other referee comments, we improved the manuscript trying to emphasize the novel findings, such as that found that the sample met the minimum levels of physical activity (150 min/week of moderate/vigorous physical activity). A native English speaker has reviewed the language of the manuscript and the other concern have been included in the limitations section. Moreover, considering the lack of similar analyses in our country, we think that our investigation may be helpful in order to identify the factors to which PA promotion interventions should be addressed.

Reviewer 3 Report

See attached.

Author Response

Reviewer 3

This study assessed physical activity levels in 383 older adults living in a specific metropolitan area in Southern Italy. In addition to describing the physical activity level in this sample, the authors sought to determine which variables might be determinants of physical activity in this population. The study design is straightforward and well structured. The results of the study, surprisingly found that the majority of the participants met the minimum levels of physical activity (150 min/wk of moderate/vigorous physical activity).

We are very grateful to Reviewer 3 for his kind judgement. We have tried to follow his suggestions in order to improve the quality of the paper.

Major concerns:

The majority of the major concerns I have could likely be addressed by editing the English language. For example, the authors refer to the sample as being random, but then describe a purposeful/convenient selection based on attendance at parishes, recreational, and cultural associations.

The authors use the term anonymous which is possible if the questionnaire is mailed or even emailed to the participants, but then use the term interviewee which indicates that they are interviewing the participants in person and therefore the participant would not be anonymous. Again, this may be a language issue, but should be corrected.

It is right. The previous version of the manuscript reported a lot of mistakes and generated misunderstanding due to an incorrect English translation. The paper has been reviewed by a native English speaker. In particular, the two above mentioned sentence were clarified. We hope that now it could be more clear.

The introduction should include some references related to the variable chosen to include as potential determinants of physical activity participation.

We have added further references regarding the factors we analyzed to explain why we included them in the study (Souza, A.M.R.; Fillenbaum, G.G.; Blay, S.L. Prevalence and Correlates of Physical Inactivity among Older Adults in Rio Grande do Sul, Brazil. PLoS ONE 2015, 10, e0117060.; Ahangar, A.A., Khoshmanzar, H., Heidari, B. Bijani A., Hosseini R., Gholinia H., Saadat P., Babaei M. Prevalence and the Determinants of Physical Activity in an Elderly Cohort of 60 years and more. A Cross-Sectional Case-Control Study. Ageing Int 2019, 44, 399–410.; Koeneman, M.A., Verheijden, M.W., Chinapaw, M.J., Hopman-Rock, M. Determinants of physical activity and exercise in healthy older adults: A systematic review. Int J Behav Nutr Phys Act 2011, 8, 142.; Ku, L.J., Stearns, S.C., Van Houtven, C.H., Lee, S.Y., Dilworth-Anderson, P., Konrad, T.R. Impact of caring for grandchildren on the health of grandparents in Taiwan. J Gerontol B Psychol Sci Soc Sci 2013, 68(6), 1009-21.)

Older adults is the generally accepted term to refer to this population, avoid using “elderly” or “old”.

The terms have been changed throughout the text.

Minor concerns/suggestions:

I’m not sure what you mean by the term “nephews”.

It is right. The previous incorrect English translation led us to report “care of nephews” instead of “care of grandchildren”, which was the issue we wanted to explore. The term has been changed throughout the text.

Many English edits are required throughout the document.

A native English speaker has reviewed the language of the manuscript.

Round 2

Reviewer 1 Report

no further comments

Reviewer 2 Report

Thank you for the responses. 

Reviewer 3 Report

This revision is much improved over the original submission - it reads a lot more clearly.  There are still a few edits throughout the document (i.e. misspelling of the word "owning" on line 248).